# Contact Tracing: Ensuring Privacy and Security

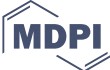

**Daan Storm van Leeuwen** [1,†], **Ali Ahmed** [2,*,‡] , **Craig Watterson** [2] **and Nilufar Baghaei** [3,‡]

1    Department of Computer Science, University of Liverpool, Liverpool 11341, UK; daanstormvanleeuwen@me.com
2    School of Engineering and Computer Science, Victoria University of Wellington, PO Box 600, Wellington 6140, New Zealand; Craig.Watterson@vuw.ac.nz
3    School of Natural and Computational Sciences, Massey University, PO Box 756, Wellington 6140, New Zealand; N.Baghaei@massey.ac.nz
*    Correspondence: ali.ahmed@vuw.ac.nz
†    Current address: Mathematical Sciences Building Campus, Albany, NY 12207, USA.
‡    These authors contributed equally to this work.

**Abstract:** Faced with the biggest virus outbreak in a century, world governments at the start of 2020 took unprecedented measures to protect their healthcare systems from being overwhelmed in the light of the COVID-19 pandemic. International travel was halted and lockdowns were imposed. Many nations adopted measures to stop the transmission of the virus, such as imposing the wearing of face masks, social distancing, and limits on social gatherings. Technology was quickly developed for mobile phones, allowing governments to track people's movements concerning locations of the virus (both people and places). These are called contact tracing applications. Contact tracing applications raise serious privacy and security concerns. Within Europe, two systems evolved: a centralised system, which calculates risk on a central server, and a decentralised system, which calculates risk on the users' handset. This study examined both systems from a threat perspective to design a framework that enables privacy and security for contact tracing applications. Such a framework is helpful for App developers. The study found that even though both systems comply with the General Data Protection Regulation (GDPR), Europe's privacy legislation, the centralised system suffers from severe risks against the threats identified. Experiments, research, and reviews tested the decentralised system in various settings but found that it performs better but still suffers from inherent shortcomings. User tracking and re-identification are possible, especially when users report themselves as infected. Based on these data, the study identified and validated a framework that enables privacy and security. The study also found that the current implementations using the decentralised Google/Apple API do not comply with the framework.

**Keywords:** contact tracing; COVID-19 pandemic; security; privacy; mobile application

## 1. Introduction

The COVID-19 pandemic has had a dramatic impact on the world, affecting millions of people. With an increasing death toll and increased COVID variants, nations are desperately investigating ways to combat the virus [1]. However, unlike, for instance, the 1918 Spanish flu, technology is playing an essential role in the fight against the virus. It comes as no surprise that authorities have embraced new, promising, and previously unavailable technology. For instance, some cities use location and movement data to assess the population's mobility, which, in turn, is an indicator of the spread of the virus. These ventures are not without scepticism–Google's flu-tracking project famously failed and showed that some techniques are not yet mature [2,3].

With regards to COVID-19, three problems make the traditional approach difficult, such as Google's flu-tracking project, along with other manual approaches, if not impossible. Firstly, contact tracing is more complicated in urban areas. A key characteristic of urbanisation is that many people live, work, and socialise in close proximity to each

other. Therefore, it is inevitable that people do not necessarily know the other people on the bus, train, gym, or marketplace. This realisation presents a challenge for a contact tracer: how do you perform contact tracing when the subject does not know most of the people they had contact with? The second problem has to do with the number of mild and asymptomatic cases that makes detection hard. The third and final problem has to do with the incubation time. The mean incubation time of 6.7 days, meaning that the time between infection and symptoms is, on average, a week [4]. This figure creates a challenge for contact tracers: when they discover a case, they might have to retrace the patients' steps for more than a week. Retracing steps becomes increasingly challenging if a patient has to rely on memory alone. The aforementioned problems combined form a deadly mix; four out of five patients exhibit no symptoms and might not even show signs for up to a week after infection—possibly longer. During that time, they are infectious to others, yet do not know they carry the disease while participating in social life in bars, public transport, and other large gatherings. If and when the condition is finally detected, manual contact tracing is extremely hard to perform. Meanwhile, for diseases such as COVID-19, with a high reproduction but low detection rate, rapid contact tracing is vital to keep cases low and the impact on society minimum [5].

Some countries were successful in their response to COVID-19. Singapore, South Korea, and China (after the initial first wave) once received high praise for their interventions. Notably, their contact tracing efforts were effective, despite the problems mentioned above (The Singapore contact tracer app also suffered from data leakage https://www.bbc.com/news/world-asia-55541001 accessed on 22 September 2021). Their approach included the use of technology, such as mobility data gathered by smartphones, giving contact tracers access to high-quality data of a person's movements before their infection. These countries were successful, as previous epidemics, such as SARS and MERS, taught them valuable lessons in handling an infectious disease [5]. South Korea even has legislation to facilitate contact tracing. With these results in mind, many states focus on digital contact tracing. Realising the power of contact tracing and the problems that come with manual contact tracing during the pandemic, governments quickly implemented digital contact tracing solutions.

The main research objectives of this paper are shown in Table 1 along with listed research methodologies to achieve them.

**Table 1.** The used research methodologies.

| Objectives | Methodology | Method |
|---|---|---|
| Threats Investigation | Qualitative | Literature Analysis |
| Investigating Available Techniques | Qualitative | Literature Analysis |
| Privacy definition for contact tracing apps | Qualitative | Literature Analysis |
| Security definition for contact tracing apps | Qualitative | Literature Analysis |
| Framework Design | Prototyping | Implementation |
| Framework Review | Quantitative | Experimentation |
| Framework validation | Prototyping | Implementation |

## 2. Digital Contact Tracing: What Is Missing?

The idea of using technology in epidemiology is not unique to the 2020 pandemic. The use of algorithms, data, and computational power to revolutionise epidemiology has been argued previously [6]. However, it was not until the publication of the work in [7] that digital contact tracing received significant attention. The authors modelled the growth of COVID-19 to assess the effectiveness of digital contact tracing. They found that the problem of pre-asymptomatic and asymptomatic cases alone is enough to sustain the exponential growth of COVID-19. If the delay between finding new cases and notification of contacts is three days or less, epidemic control is possible. Moreover, immediate notification by

an application could lead to epidemic control if user adoption is 56% or higher. Lower user adoption can still contribute significantly to halting the exponential spread of the epidemic, when combined with additional interventions, such as face masks, frequent hand sanitation, and social distancing. "Digital contact tracing could play a critical role in avoiding or leaving lockdown" [7]. It is worth noting that two countries, amongst many, that successfully employed digital contact tracing solutions are China and South Korea. However, whether their methods respect privacy and security are debatable. For example, China ties its social gigantic application WeChat (https://www.wechat.com accessed on 22 September 2021) to various other governmental IT systems. Realising this threat, privacy and security advocates worldwide quickly warned of the dangers of digital contact tracing. As early as April 2020, an impressive global community of scientists released a press statement calling for digital solutions that respect privacy and security [8].

Even with the debate on privacy and security still raging, governments decided to develop contact tracing applications. Most, but not all, based their apps on the Google and Apple exposure notification (GAEN) framework. This framework is, in essence, an extra layer between the operating systems of both Google (i.e., Android) and Apple (i.e., iOS) and a third-party contact tracing application. GAEN uses Bluetooth low energy (BLE) hardware for proximity tracing. GAEN is essentially based on the decentralised DP-3T protocol and promises a privacy-preserving method to allow digital contact tracing. However, some governments remain unconvinced. The French government specifically decided to use its proprietary implementation, not wanting to rely on Apple and Google. It is argued that GAEN is inherently dangerous especially against a malicious authority [9] and is considered too restricted to perform digital contact tracing effectively. It proposes to use more, instead of less, personal information [10]. Although governments are rushing to develop contact tracing applications, often based on Google and Apple exposure notification (GAEN), the fundamental questions of privacy and security are debated and remain unanswered to this day. Although the argument continues, the question remains: what system ensures privacy and security best? This observation is troublesome considering that there are guidelines and frameworks for many other applications to ensure that those apps are safe to use, but this is lacking for digital contact tracing. In essence, what framework should app developers, governments, and healthcare providers follow when deploying a digital contact tracing system? It is worth emphasising that decentralised contact tracing apps have inherent privacy limitations [11]. It is worth noting "the Google Play Services component of these apps is problematic from a privacy viewpoint" [12] (i.e., mainly for GAEN-based contact tracing apps users on Android). This may justify why many people are reluctant to download and use contact tracing apps (i.e., The health-privacy trade-off [13]).

Known as test, trace, isolate, and quarantine (TTIQ), this strategy isolates only at-risk patients, instead of the entire population. This approach requires finding cases early, which, in the case of COVID-19, is difficult, as discussed earlier. Contact tracers in this situation require help to identify possible cases. Fuelled by the initial success that China, South Korea, and Singapore had with contact tracing [14], dozens of nations are now considering digital contact tracing solutions to identify cases quickly because "app-based tracing remains more effective than conventional contact tracing" [15].

Contact tracing consists of two components or systems: an epidemiological component, which is not the scope of this project, and a technical component [16]. Both have to work together for the systems to be effective [17]. Since smartphone penetration reaches 80% in the European Union [18], most attention concerning the technical component goes to smartphones. Contract tracing applications are mostly using four technologies or sensors on smartphones [5]:

1. Cell. Using the mobile operators cell tower information. Crude, but simple;
2. Wi-Fi. Scan for nearby devices connected to the same Wi-Fi network;
3. GPS. Using GPS and other positional data. However, these data, too, are relatively crude [19], especially in urban areas;

4.　Bluetooth. Bluetooth beacons allow for short range transmission and reception and includes signal strength—specifically Bluetooth low energy (BLE).

Discussing what technology to use goes hand in hand with the question: does that technology help curb the infection, and find the effective growth rate $R_e \leq 1$? Research shows that it does if adoption exceeds 6% [7]. However, even if user adoption is significantly lower, digital contact tracing applications can have a significant contribution to stop epidemic growth—especially when combined with other measures [7,15,16]. However, speed is of the essence: the time between detection of a case and notification of contacts must be as quick as possible: "Combining traditional and digital contact tracing may leverage the advantages, and mitigate the limitations, of each approach" [17]. Given these requirements, BLE proved the best sensor to estimate the relative distance between two people—even though BLE suffers from false positives [5]. However, recording proximity between different people inherently leads to privacy and security questions [17].

## 3. Background Discussion

As soon as digital contact tracing surfaced, researchers began debating the privacy and security implications. Within Europe, two systems emerged from this debate: centralised and decentralised which are in the focus of this section. The first serious proposal in Europe came from the Pan European Privacy-Preserving Proximity Tracing (PEPP-PT) group (https://ercim-news.ercim.eu/en121/ib/pan-european-privacy-preserving-proximity-tracing accessed on 22 September 2021). They proposed a protocol, also named PEPP-PT, which works with BLE to record chance encounters between random individuals. The protocol functions by storing these encounters on the handset of the user. If the user receives a COVID-19 diagnosis, they can choose whether to upload the encounter data to a central server. The central server processes the encounters and notifies other, at-risk users by either a push or pull mechanism (PEPP-PT, 2020). The fact that it is the server that processes risks makes this system centralised. The French ROBERT has its origins in PEPP-PT, which drew many privacy concerns. This is why a decentralised system, where all processing takes place on the handset itself was proposed (e.g., the decentralised privacy-preserving proximity tracing (DP-3T) protocol). Unlike PEPP-PT, DP3T proposes that all infection data are publicly available and pushed to all handsets thus that every phone can assess the risk locally instead of centrally. The decentralised approach that the DP-3T-team proposed, inspired the exposure notification framework that Google and Apple jointly developed.

PEPP-PT attracted a lot of negative attention from scientists worldwide. On 19 April 2020, more than 400 scientists and researchers signed a Joint Statement on Contact Tracing, explaining a fear for mission creep in contact tracing systems stating that "It is vital that, in coming out of the current crisis, we do not create a tool that enables large scale data collection on the population" [8]. This fear is rooted in the fact that systems that process data centrally can, in theory, also produce a social graph: a detailed map that shows who has been in contact with whom. Opposed to that idea, the authors propose a decentralised system, where all processing takes place on the handset itself (e.g., the decentralised privacy-preserving proximity tracing (DP-3T) protocol). Unlike PEPP-PT, DP3T proposes that all infection data are publicly available and pushed to all handsets thus that every phone can assess the risk locally instead of centrally. The decentralised approach that the DP-3T-team proposed, inspired the exposure notification framework that Google and Apple jointly developed.

The DP-3T design forms the basis for the Google and Apple exposure notification (GAEN) API. The systems differ from PEPP-PT in that it reveals minimal information to the backend server. Where PEPP-PT calculates risk information in the backend and notifies the users at risk, DP-3T pushes at-risk information to all handsets, relying on each handset to calculate risk information and alert the user—making it decentralised. This means the user does not need to register for the service. The DP-3T protocol follows two steps. Firstly, generation and storage of ephemeral IDs. This handset generates a daily rotating key $SK_t$

and uses a hashing algorithm to derive the EphIDs. These are broadcast to other handsets. It is impossible to recalculate the original $SK_t$ based on the received EphIDs. Secondly, proximity tracing. Whenever a user reports positive for COVID-19, the healthcare authority publishes the $SK_t$ for the days the user was infectious. Other phones can download and use these to re-calculate the EphIDs and compare them to the ones stored in memory.

Contact tracing applications fall within the scope of the GDPR [20]. The European Data Protection Board (EDPB) investigating both systems and confirmed they can be compliant to the GDPR, with an interesting footnote: the EDPB assesses the decentralised system better in line with the data minimisation principle. A decentralised system is preferred and it is argued that "Public health bodies are, at least in theory, more democratically accountable. On the other hand, users have, at least in theory, more robust rights to withdraw from commercial systems operating based on user consent" [21]. Even though those positions give direction to what systems to use, national privacy authorities decide if an app is admissible. For the centralised PEPP-PT/ROBERT, the French privacy authority CNIL cautiously approved the French governmental Stop-Covid application (https://www.cnil.fr/fr/publication-de-lavis-de-la-cnil-sur-le-projet-dapplication-mobile-stopcovid accessed on 22 September 2021). The day after this approval, however, a group of 471 French security and cryptography researchers released a statement warning of the danger of digital contact tracing, regardless of which system ( https://uk.news.yahoo.com/hundreds-french-academics-sign-letter-155630916.html accessed on 22 September 2021). A few days later, on 26 April, Germany withdrew from the ROBERT framework, opting for a decentralised approach instead. However, Germany did so because of the earlier mentioned GAEN-advantage that allows for continuous background access to the Bluetooth hardware. The Netherlands, Germany, and Switzerland all received approval for their decentralised/GAEN Apps from their respective privacy authorities as well. Researchers from the University of Cambridge also looked into the decentralised (GAEN) implementation vs. the GDPR. They also conclude that GAEN is compliant with the GDPR: "the GDPR's expansive scope is not a hindrance, but rather an advantage in conditions of uncertainty such as a pandemic" [21]. However, the verdict on whether a contact tracing application complies with GDPR depends upon both GAEN and the application. That does not mean the debate about GAEN ends there. Much attention goes to the role Google and Apple play and how they expand their healthcare influence. For example, the work in [22] extensively investigates GAEN and argues that the industry practice by Google and Apple to remember which user and which handset downloaded what apps (including contact tracing apps), is against the GDPR. These debates are relevant but exceed the discussion on GDPR-compliance. The German withdrawal from PEPP-PT, the cautious approval by CNIL, and the Inria-split made the introduction of PEPP-PT controversial. The split caused reputation damage to PEPP-PT, rubbing off on the perception of privacy and security. The position of European's political and privacy bodies, supported by research, all conclude that decentralised solutions are preferable over centralised ones. It seems that decentralised technology is the best way to design contact tracing apps. However, there are other positions to consider as well, beyond centralised and decentralised, such as the DESIRE protocol [23]? The DESIRE protocol which, unfortunately, has not been comprehensively studied or used in any contract tracing application yet due to the lack of protocol software libraries.

DP-3T and PEPP-PT have data minimisation at the core, which the GDPR requires. However, outside of the European Union, proposals exist that expand, rather than limit, the amount of data collected, to increase effectiveness. It is recommended to build a voluntary system that fits the needs set forth by public health authorities. Additionally, it allows a user to enable or disable additional sensors or contextual information they want to use within the application [24]. These recommendations place a lot of trust and responsibility in the hands of the user. In other words, let the user decide how much data they want to share! The idea to add contextual information in contact tracing applications begs the question of what this would do to privacy and security. The use of contextual

information in South Korea is examined and it was found that people could be the subject of "Stereotyping, stigmatisation and discrimination" [25]. It is for that, and other reasons that the GDPR calls for data minimisation and requires that actors take proportionality (is the expected gain worth it versus the level of breach?), subsidiarity (can the effect be reached by another, less intrusive method?), and necessity (is the breach necessary?) into account. From that perspective, enriching data is not a good idea since people could feel compelled or pressured to release more data than they prefer. In addition, if people want to release additional data to facilitate contact tracing, they can already do so. Typical methods could include browsing their social media history, checking their calendar, or reviewing navigation data.

With the research identified in this section, and the importance various authors placed on various aspects, it is possible to come up with a list of requirements for contact tracing applications–a framework. However, two problems remain: First, some requirements find their origin in untested research. Second, if the decentralised Google and Apple API is the best option for contact tracing, will it survive that framework? Does the Google and Apple implementation comply with the research identified so far? Additionally, the role Google and Apple play is debatable. What are they going to do with the technology they designed once the crisis is over? The same question needs answering from a governmental perspective: Somewhere, a policymaker in law enforcement already considers using this technology for law enforcement purposes.

Since contact tracing is dealing with sensitive information, this information needs to be protected against well-known attacks. Understanding the attack vector against such applications is important in securing them. Investigating the threats against both centralised and decentralised systems are demonstrated in [9,26]. It was argued that neither system protects privacy and security [9] (i.e., tracking, social graphing, identification, pressure to opt-in, replay attacks, etc.), as shown in Figure 1. Some of the threats against decentralised systems include tracking people by de-pseudonymising user's EphID, disclosing the social graph, identifying diagnosed people, pressure to opt-in, and injecting false encounters. Centralised systems suffer from similar threats.

A plus (+) indicates this schema is preferred; a minus (-) indicates this schema is not preferred.

| | Centralised | Decentralised |
|---|---|---|
| **Tracking** | (-) Central server puts all users at risk; possibly theft of on device-data | (+) Infected users at risk; possibly theft of on-device data. |
| **Social graph** | (-) Central server puts the social subgraph at risk | (+) Not possible with the possible exceptation of RPI theft |
| **Identification (de-pseudonymization)** | (-) Possible by RPI reception & observations; aggrevated by central server attack. | (+) Possible by RPI reception & observations. |
| **Identification (diagnosed people)** | (+) Difficult; if not impossible. | (-) Identification diagnosed people is possible. Legislation could mitigate this attack. |
| **Pressure to opt-in** | (-) Pressured user always results in privacy risk. | (+) Pressured user retains option to opt-in if the health app is designed with an OTK authorisation. |
| **Replay attacks** | (+) Replay of RPIs is possible, less opportunity for dark markets, effect measurement still possible but more labour intensive | (-) Replay of RPIs is possible, with a dark market for high likelihood infectious RPIs as a possibility. This makes it possible to measure the effect of the replay attack. |

**Figure 1.** Overview of privacy and security attacks.

## 4. Framework Analysis and Design

In this section, we will introduce the framework we built and tested against several leading European implementations. Based on the comprehensive literature survey, the draft framework that preserves privacy and protects the security of users is demonstrated in Table 2. The table provides information on and an overview of requirements that have to be met to make an app that is both private and secure within the European Union. Notice that a decentralised design is required, in line with the EU and research. This (draft) framework in itself satisfies objective five, but it is unknown whether the current, leading implementations meet this framework and these requirements.

**Table 2.** Draft framework for privacy and security.

| Identifier, | Requirement Number, | Objective |
|:---:|:---:|:---:|
| 1.1 | Users cannot be tracked | 1 (Threat) |
| 1.2<br><br>Exposure Keys (TEKs) | Users cannot be re-identified either after infection or otherwise. This includes when uploading infectious temporarily<br>1 (Threat) | |
| 1.3 | Users cannot be pressured into disclosing sensitive data | 1 (Threat) |
| 1.4 | Replay attacks leading to false alerts are impossible | 1 (Threat) |
| 1.5 | Users cannot be identified through profiling or application use | 1 ( Threat) |
| 2.1 | Proximity tracing is exclusively performed with BLE | 2 (Techniques) |
| 2.2 | Decentralised design conforming with DP-3T/GAEN | 2 (Techniques) |
| 2.3 | The BLE beacons are pseudonymous and rotates frequently. This includes fake hardware addresses. | 2 (Techniques) |
| 3.1 | Every implementation is approved by the privacy authority with a valid PIA published. | 3 (Privacy) |
| 3.2 | The contract tracing system is voluntary and dismantled after the pandemic preventing mission creep. | 3 (Privacy) |
| 3.3 | Google and Apple dismantle the GAEL-API after the pandemic | 3 (Privacy) |
| 3.4 | The backend stores data exclusively in the EU with no data transfers outside the EU. | 3 (Privacy) |
| 4.1 | The app follows a security standard to ensure continuous security evaluation | 4 (Security) |
| 4.2 | The app source code is open and secured against malicious updates | 4 (Security) |
| 4.3 | Google and Apple disclose the GAEN API | 4 ( Security) |

To validate the draft framework, the project tested several leading European implementations against it. The results gathered from this paper can verify whether the GAEN-enabled implementations comply with the framework and are, as such, ensuring privacy and security. The requirements presented later, translate to the questions in Table 3

that direct further research. Note that requirement 2.2 is not tested and considered one of the future works. This requirement stipulates a decentralised design, which is an integral part of the experiments shown later. The research focuses only on decentralised designs. It is worth noting that Table 3 identifies three different research methods to investigate the answers to the questions.

**Table 3.** Questions to evaluate implementations.

| Research Question, | Identifier, | Method |
|---|---|---|
| Can infected users be re-identified on uploading TEKs by sniffing their traffic? | 1.2 | Review Implementation |
| Can users be identified by IPs on uploading TEKs? | 1.2 | Review Implementation |
| Is Bluetooth the only sensor used in the application? | 2.1 | Review Implementation |
| Does the implementation have a PIA approved by the national privacy authority? | 3.1 | Review Implementation |
| Is the application voluntary? | 3.2 | Review Implementation |
| Does the backend store data exclusively in the EU? | 3.4 | Review Implementation |
| Does the implementation follow a security standard ensuring continuous security assessment? | 4.1 | Review Implementation |
| Can the source code be reviewed? | 4.2 | Review Implementation |
| Does the downloadable app match the source code published? | 4.2 | Review Implementation |
| Can GAEN API be reviewed? | 4.3 | Review Implementation |
| Are reply attacks possible? | 1.4 | Literature Review |
| Did the government have plans to dismantle the system after the pandemic? | 3.2 | Literature Review |
| Do Google and Apple have plans to dismantle GAEN after the pandemic? | 3.3 | Literature Review |
| Can users be tracked by Rolling Proximity Identifier (RPI)? | 1.1 | Experiments and Literature Review |
| Can users be re-identified by RPIs analysis? | 1.2 | Experiments |
| Can users be pressured into disclosing compromising data? | 1.3 | Experiments |
| Are the BLE beacons pseudonymous and do they rotate frequently? | 2.3 | Experiments |
| Are Bluetooth hardware addresses in the advertisement data random and rotate frequently? | 2.3 | Experiments |
| Is it possible to profile users? | 1.5 | Experiments |

## 5. Framework Evaluation

It is worth noting that this research used three leading European implementations, which are not only currently in use but also were subjected to a lot of attention and debate. These implementations are CoronaMelder (i.e., the Netherlands), Corona-Warn-App (i.e., Germany), and SwissCovid (i.e., Switzerland). These Apps are mature and well-received within the privacy research community [19]. Now, let us revisit the questions in Table 3 and

answer them. It is worth emphasising that, this paper focuses on those questions requiring experimentation (i.e., see Table 3), thus experiments were performed using a Raspberry Pi 4 running Raspbian Linux and an Ubertooth Bluetooth interception module. Different mobile handsets are used in such experiments.

*Can infected users be re-identified on uploading TEKs by sniffing their traffic?* Uploading TEKs is a sign of infection since it creates traffic to a server that is normally absent. This attack works even if traffic is encrypted. To negate this threat, the Dutch and German implementation upload fake keys at random intervals. If an attacker sniffs the traffic between the handset and the server, it is impossible to separate real uploads of keys from fake keys. This implementation prevents an attacker from identifying infected users by sniffing the TCP/IP traffic between the handset and the server. *Can users be identified by IPs on uploading TEKs?* If an attacker compromises the backend server, it is possible to re-identify infected people by IP address. This only works when it is possible to combine the IP address with infection data. The Dutch and German implementations separate the IP address from the TEKs as soon as traffic arrives at the backend server [27]. The backend stores the TEKs in a database, but it stores the IP addresses for a maximum of 15 min to allow for intrusion and attack detection. This solution shows that it is possible to separate identifiable IP information from the pseudonymous TEKs, preventing re-identification when uploading TEKs.

*Is Bluetooth the only sensor used in the application?* The three implementations were examined on both iPhone 11 Pro (i.e., iOS) and Nokia TA-1042 (i.e., Android). All three apps ask for no other permissions than to use the exposure notifications, mobile data for downloading TEKs and the general notifications. The investigation shows that the GAEN-enabled apps of the Netherlands, Germany, and Switzerland do not require any additional authorisations.

*Does the implementation have a PIA approved by the national privacy authority?* The European Data Protection Board is very clear that countries, wishing to implement a contact tracing application, need to publish a Data Protection Impact Assessment (DPIA) and have it approved by their local privacy authority. An approved DPIA exists for the German, Dutch, and Swiss apps [27,28].

*Is the application voluntary?* In all three nations, there is no legislation in place to make the app mandatory. GDPR also prohibits this, but in the Netherlands, additional legislation was enacted specifically forbidding private or public organisations to make the app mandatory. When installing and running SwissCovid, Corona-Warn, and CoronaMelder on both Android and iOS devices, none of these apps required registration. The privacy authorities of these countries do not allow registration since this would link personal information to the app, an IP address and possibly the TEKs [27]. The only app that did require registration is the French TousAntiCovid, in line with the PEPP-PT protocol. The European Commission (2020), parliament (2020) and EDPB confirm the importance of these criteria. The app must be voluntary and should be dismantled after use. Upon examining the DPIA of the countries investigated, it is confirmed that this is the intention [27,28]. However, the work in [29–31] make a legitimate argument that the technology cannot be un-invented and that even though national governments might dismantle the app after the pandemic, Google and Apple might not. Google and Apple claim that they will only use the technology during the pandemic, but there is no way of knowing whether this intention solidifies in reality.

*Does the backend store data exclusively in the EU?* When studying SwissCovid, processing data outside the European Union is undesirable, especially after the European Court of Justice invalidated Privacy Shield in 2019 [9]. Without additional safeguards and contractual clauses, the transfer of personal information to the US is illegal. SwissCovid uses Amazon servers for the back-end. Interestingly, both the Dutch and German privacy authorities agree that the backend should not be hosted on US servers. Those applications use servers within Europe, to avoid the transfer of personal data. Based on the DPIA of the

Netherlands and Germany, and the criticism on the Swiss implementation, the backend should store data within the EU which means the criterion is valid.

*Does the implementation follow a security standard ensuring continuous security assessment?* When designing software, a security standard helps to prevent common mistakes, forgetting critical items and review the software created. Well-known examples stem from the International Standards Organisation (ISO), the National Institute for Standards and Technology (NIST) and the European Union. The Dutch and German applications follow the NIST Cybersecurity directive, to provide cyclical security assessment and enhancements.

*Can the source code be reviewed?* The EDPB and various national privacy authorities stress the fact that the source code of Corona applications must be published as open source. A popular platform to do so is GitHub. Indeed, the three Apps examined all have their source code published as follows. For CoronaMelder, source code is accessible from https://github.com/minvws/nl-covid19-notification-app-ios accessed on 22 September 2021 and https://github.com/minvws/nl-covid19-notification-app-android accessed on 22 September 2021. For CoronaWarn App, it is accessible via https://github.com/corona-warn-app/cwa-app-ios accessed on 22 September 2021 and https://github.com/corona-warn-app/cwa-app-android accessed on 22 September 2021. For SwissCovid, one can view the source code from https://github.com/DP-3T/dp3t-app-ios-ch accessed on 22 September 2021 and https://github.com/DP-3T/dp3t-app-android-ch accessed on 22 September 2021.

*Does the downloadable app match the source code published?* Even though the software designers publish the source code online, that does not mean it matches the version of the app that is available for download in the App Store and Play Store. For instance, at the time of writing, the version of CoronaMelder at GitHub is 1.0.12, while the version on our phone is 1.0.11. Thus, how can one verify the app indeed matches the one published on GitHub? The Dutch and German App developers use an external security organisation to verify this is indeed the case [27]. Generally, this criterion is therefore valid and solvable, with a proper implementation.

*Can GAEN API be reviewed?* Neither Apple nor Google released the full, implemented source code of their APIs. Google did publish a "reference design" on GitHub (https://github.com/google/exposure-notifications-android accessed on 22 September 2021, https://developer.apple.com/exposure-notification/ accessed on 22 September 2021). The objective of both releases is not to allow external parties to review the code and participate in finding flaws, but to give developers an idea of how the internals of the GAEN API work. This problem is quite apparent in the literature. Many authors point out the inability to review the design choices, security and privacy of the GAEN-API [9,27,29]. The question is if there are other possibilities to review the source code. An external party could be contracted to review the code, on behalf of the health authorities.

*Are replay attacks possible?* Given a decentralised design, the work in [19] proved that a relay-based worm-hole attack is possible, generating false contact on a target's device. They performed their experiment in reaction to the SwissCovid app. The Swiss NCSC (2020) The attack is confirmed by the work in [9,26,28,32]. An important realisation of all these articles is that the authors replayed the RPI, not the TEK. This proves that, at least in decentralised designs, replay attacks remain possible and real danger.

The following mobile handsets were used to perform the experiment:

1. iPhone 5S incompatible with GAEN and contact tracing applications. This phone serves as a comparison between devices running GAEN, as a device that cannot run GAEN;
2. A Nokia TA-1047 running Android 10, a cheap but GAEN-compatible smartphone, in developer mode;
3. A Samsung S9 Android 9 running GAEN, a typical consumer smartphone;
4. An iPhone 11 Pro running GAEN, a typical consumer smartphone;
5. A Ruggear RG-850 Android 10 phone running GAEN, a typical smartphone used with healthcare professionals.

*Are the BLE beacons pseudonymous and do they rotate frequently?* The GAEN-documentation sets the time interval for RPIs at roughly 10 min. By hooking up an Android phone in developer mode with the HCI Snoop Log enabled to a computer with Android Studio running, it is possible to filter for RPI changes.

Figure 2 shows that every 10 min, the RPI changes to a new value. Deviations were observed; sometimes the RPI changes in 7 min, sometimes in 12 min. A pattern between these times could not be determined; it is reasonable to suspect that a pseudo-random generator determines a time shift. The log also shows the new *RollingProximityID* value when the RPI rotates. These values appear random; no correlation between the values was observed. The question is whether these RPIs match the ones transmitted by the phone. A single GAEN-equipped phone with the Dutch CoronaMelder GAEN application was placed next to the interception station to investigate this. The purpose of this test is to compare the transmitted RPI with the received RPI. The Ubertooth intercepted the BLE broadcasts and saved them to a *pcapng* file for later analysis. Using Wireshark for analysis, the following filter ensured that only GAEN BLE advertisements show up, which are close by ($\geq$45 dbm). Figure 3 demonstrates the output after capturing the data.

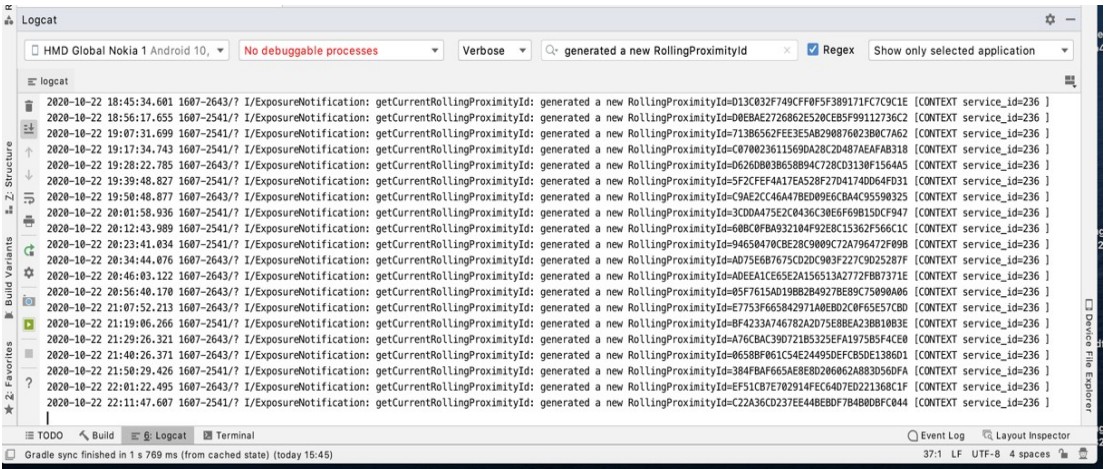

**Figure 2.** Time interval of RPI change.

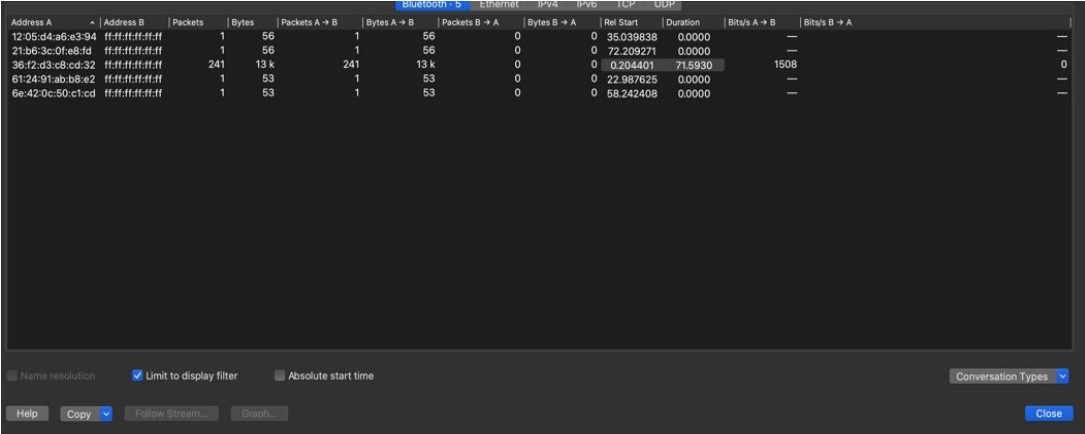

**Figure 3.** Wireshark Conversation.

Almost all packets were transmitted by *BD_ADDR 36:f2:d3:c8:cd:32*. The four packets that were transmitted from other addresses, turned out to have a faulty cyclic redundancy check (CRC), a characteristic of an incomplete interception. All other packets showed the same RPI transmitted, with exception of the AEM as shown in Figure 4. The received RPI received *f18cf5f57aff05fde6779a562cc413bd* matches the one transmitted by the telephone, as seen in the HCI Snoop Log. The experiment proved that the reported RPI in the HCI

Snoop Log indeed matches with the RPI transmitted by the phone, and subsequently received by the Ubertooth. This experiment confirms the RPIs rotate frequently and are indeed pseudonymous.

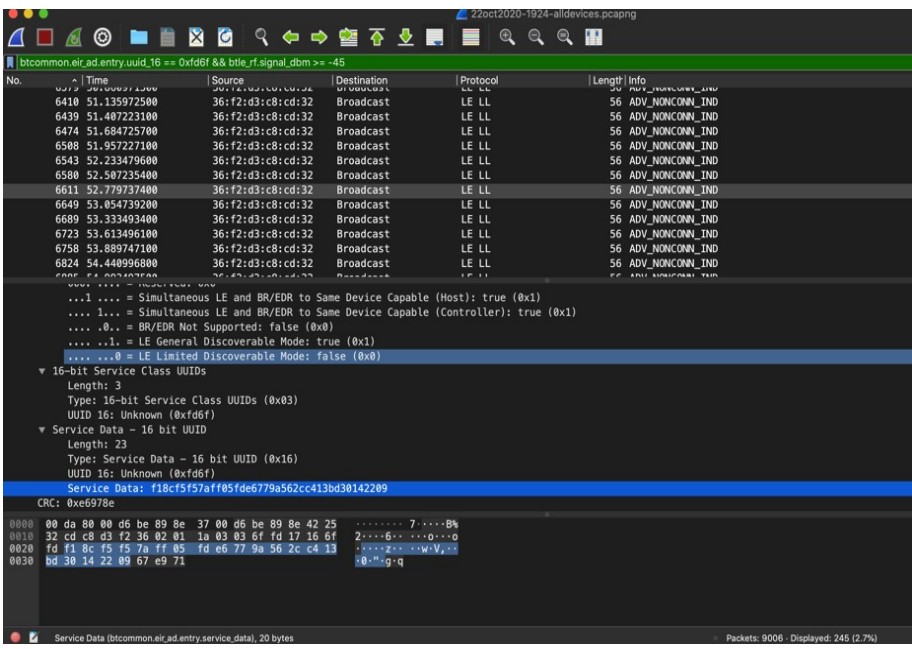

**Figure 4.** The Received RPI of a random package.

*Are Bluetooth hardware addresses in the advertisement data random and rotate frequently?* The Google and Apple specification requires the MAC address to rotate frequently (Google, 2020), to avoid device tracking. This experiment checks to see if indeed the MAC changes. The results were analysed with Wireshark below. As Figure 5 shows, address *77:8c:e9:ca:9c:55* first started transmitting 0.144961s after interception started and stopped 193.8150 s later (193,959961 s after experiment start). Interestingly, 194.498694 s after experiment start, *55:7d:89:91:fc:d2* started transmitting. Could this be a MAC address change for the same device? To answer this, one could see in Figure 6 the RPI that the first address used.

**Figure 5.** The Wireshark addresses captured.

**Figure 6.** RPI transmitted by 77:8C:E9:CA:9C:55.

Wireshark can filter on this RPI, which shows that the new address transmitted the same RPI as seen in Figure 7.

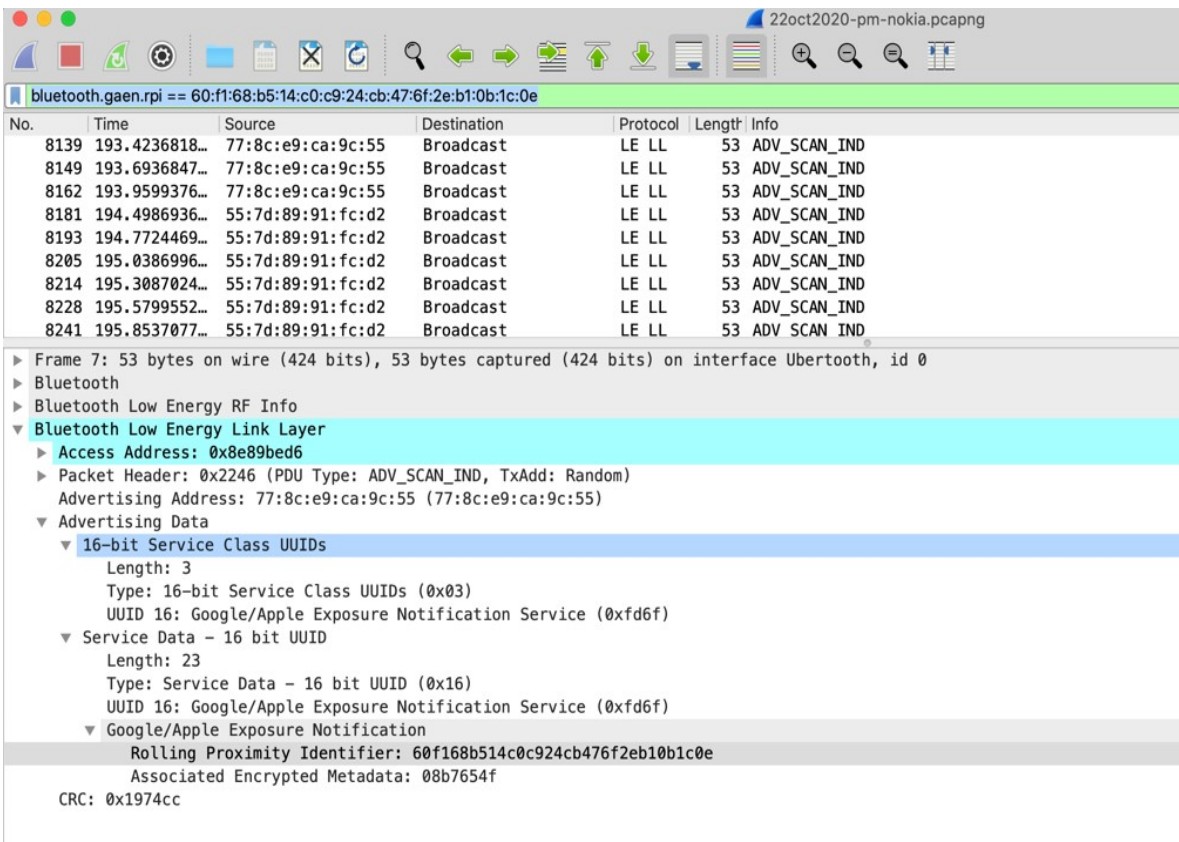

**Figure 7.** MAC address change with same RPI.

This experiment proves that even though GAEN randomises the MAC addresses frequently, it is possible to track the same device by using the RPI. Since RPIs rotate roughly every 10 min, and an RPI change forces a MAC address change as well, this means that an attacker has a maximum of 10 min to track a device. Even though the MAC address changes, a relatively easy analysis can tie the new MAC address to the old, by using the RPI.

*Can users be tracked by RPI?* Harvesting RPIs is not difficult using tools such as Raspberry Pi or Wireshark. Various projects on GitHub do exactly this: collecting RPIs. One such example is Corona-Teller (https://github.com/zeno4ever/CoronaTeller accessed on 22 September 2021). An attacker can integrate other data, such as location, date, and time, without any effort, and upload it to a central repository. This clearly shows that RPI harvesting is not difficult, and easily automated. Once harvested, it is possible to track users when they report an infection. The work in [28] proved this attack. Using the conclusions from the experiments performed above, it is possible to confirm these results: Intercepting RPIs is relatively easy and the attacker can combine them with location data, time, and date. By uploading these results to a central database, it is possible to track a user once infected.

*Can users be re-identified by RPIs analysis?* As discussed earlier, even though the MAC address changes frequently, it is possible to follow a specific device for a maximum of 10 min in which it transmits and receives the same RPI. This begs the question: is it possible to re-identify a user in that period? It is possible to combine phone transmissions with visual observations, as described in [9,26]. However, this experiment found a digital possibility as well. In this experiment, an attacker observes a user relatively close by and notices the user forms a Bluetooth piconet with for instance a headset, smartwatch, or

another device. The attack starts intercepting Bluetooth low energy and notices the victim uses GAEN. To filter out other GAEN-users, the attacker applies a filter so only relatively close devices are shown in Figure 8.

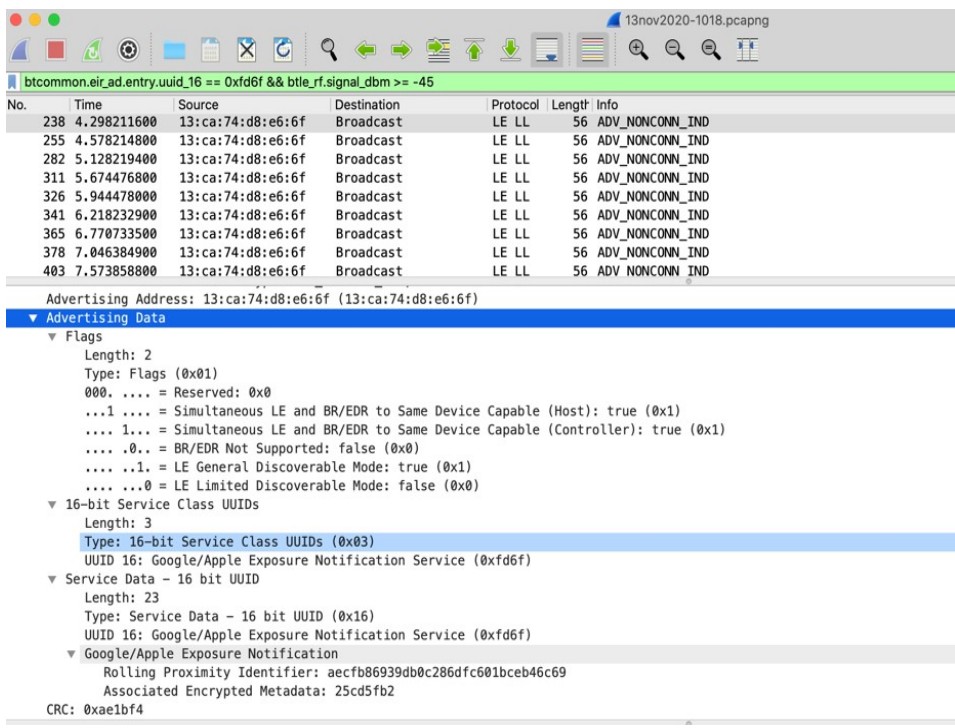

**Figure 8.** Victim uses GAEN; displays RPI.

The attacker can now visually link the user to the RPI, but the MAC address displayed is random. The attacker is not able to retrieve further information using this MAC. The piconet does display (part of the) MAC address, but only lower address part (LAP). To query a device, the attacker needs both the upper address part (UAP) and LAP. The attack now employs the Ubertooth to scan for networks and follow networks to try to retrieve the UAP. The attacker does that twice. The scan can retrieve the UAP and query the device, as shown in Figure 9. The figure shows, the Ubertooth successfully retrieved the UAP of the device and was able to query it. The UAP *79:93:BF:F9* matches the phone's address used in the experiment. The attack now successfully retrieved the RPI, identified the user and found the significant part of the Bluetooth address. The attacker can escalate by performing, for instance, a Bluetooth bias attack, impersonating an already paired device to retrieve more compromising information [33].

A serious consideration with this attack is that there are three limitations for it to work. First, the attack only works in a relatively closed setting, where visual observations confirm that no other devices are present. Second, the victim must use a Bluetooth piconet. Third, the attack is relatively labour-intensive.

*Can users be pressured into disclosing compromising data?* TEK (or $SK_t$) is the most compromising information. With the TEK, an attacker can reconstruct all RPIs. According to the documentation that Google and Apple provide on GAEN, it is impossible to retrieve this information since the devices store it in an enclave, or protected part of the memory. When investigating the exposure checks on both iOS and Android, the devices reveal no sensitive information. The user can check when the device downloaded TEKs, what the hash of the TEK download is, how many keys are in the file and how many match, etc. The user also has the option to delete the keys with the "Delete Exposure Log" button. However, the user cannot access sensitive information.

```
Scan results:
??:??:??:2B:28:0F
AFH map: 0xefd0023df80f01000011
??:??:79:93:BF:F9          iPhone-11-Pro-Daan
Requesting information ...
        BD Address:  00:00:79:93:BF:F9
        Device Name: iPhone-11-Pro-Daan
        LMP Version:  (0xa) LMP Subversion: 0x4228
        Manufacturer: Broadcom Corporation (15)
        Features page 0: 0xbf 0xfe 0xcf 0xfe 0xdb 0xff 0x7b 0x87
                <3-slot packets> <5-slot packets> <encryption> <slot offset>
                <timing accuracy> <role switch> <sniff mode> <RSSI>
                <channel quality> <SCO link> <HV2 packets> <HV3 packets>
                <u-law log> <A-law log> <CVSD> <paging scheme> <power control>
                <transparent SCO> <broadcast encrypt> <EDR ACL 2 Mbps>
                <EDR ACL 3 Mbps> <enhanced iscan> <interlaced iscan>
                <interlaced pscan> <inquiry with RSSI> <extended SCO>
                <EV4 packets> <EV5 packets> <AFH cap. slave>
                <AFH class. slave> <LE support> <3-slot EDR ACL>
                <5-slot EDR ACL> <sniff subrating> <pause encryption>
                <AFH cap. master> <AFH class. master> <EDR eSCO 2 Mbps>
                <EDR eSCO 3 Mbps> <3-slot EDR eSCO> <extended inquiry>
                <LE and BR/EDR> <simple pairing> <encapsulated PDU>
                <err. data report> <non-flush flag> <LSTO> <inquiry TX power>
                <EPC> <extended features>
        Features page 1: 0x0f 0x00 0x00 0x00 0x00 0x00 0x00 0x00
        Features page 2: 0x7f 0x07 0x00 0x00 0x00 0x00 0x00 0x00
        Clock offset: 0x2b5d
        AFH Map: 0xfefffffeffffffffff3f
AFH map: 0x04000000000150640101
```

**Figure 9.** Retrieved UAP using the Ubertooth.

## 6. Conclusions and Future Work

Can contact tracing applications, deployed in the fight against COVID-19, be safe and secure and, if so, what framework should designers follow to reach safety and security? This question started this research project. This paper investigated the current literature that exists on contact tracing from security and privacy perspectives. It led to the conclusion that decentralised solutions are preferable to centralised solutions. Decentralised solutions provide less attack surface because of the distributed nature of the system. The paper proposed a framework that provides a roadmap on building contact tracing applications within the EU. The framework is evaluated against the threats identified earlier using three leading European contact-tracing implementations. The results proved that the framework is valid and provides safety and security. However, the results also showed that the decentralised principle has inherent properties that lead to a breach in privacy and security.

This study intended to research and validate a framework for mobile contact tracing applications in light of the COVID-19 pandemic. The application of the research is not just limited to COVID-19, but any infectious disease. However, the research is limited in that it looked more thoroughly at decentralised than centralised solutions. Even though this study provided a comprehensive overview of contact tracing techniques and applications, follow-on work could look at centralised systems in more detail. Another suggestion for follow-on research is to develop an application that conforms to the framework: when starting from scratch, without the work that Apple and Google did, is it possible to design an application that conforms to the framework? A possible solution is to look at hybrid systems, which combine the best of centralised and decentralised systems.

**Author Contributions:** Conceptualization, A.A.; formal analysis, D.S.v.L.; investigation, D.S.v.L.; writing—original draft, D.S.v.L.; methodology, A.A.; software, D.S.v.L.; supervision, A.A.; validation, A.A., C.W. and N.B.; writing—review and editing, A.A., C.W. and N.B. All authors have read and agreed to the published version of the manuscript.

**Funding:** This research received no external funding.

**Institutional Review Board Statement:** Not applicable.

**Informed Consent Statement:** Not applicable.

**Conflicts of Interest:** The authors declare no conflict of interest.

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
