# Peer review of "Contact Tracing: Ensuring Privacy and Security"

_applsci, doi:10.3390/app11219977_

Round 1

Reviewer 1 Report

The article is overall well written, the contributions are well presented, the references are numerous and striking, the experiments are well conducted.

Minor remarks: 

In the introduction, the authors should talk about the Singapore contact tracing application, which is not very privacy-friendly:
https://www.bbc.com/news/world-asia-55541001

The sentences at the top of page 4 seem to be repetitions of what has already been written at the bottom of page 3.

I think all the figures should be edited by removing at least the decorations, to focus only on the essential.
Page 9, at the bottom of the page, there is an "8." What is it?

The conclusion highlighting the advantages of the decentralized version is a bit lapidary and lacks justification in light of everything else written in the article. A summary of the attacks in a table would be welcome.

The last sentence should be qualified with respect to reference [23] which is a combination of centralized and decentralized contact tracing. 

Author Response

Reply to Reviewer 1:

Firstly, we appreciate the constructive feedback. We have done our best to accommodate them into the manuscript and those we could not, we have provided a reason/explanation.

  1. In the introduction, the authors should talk about the Singapore contact tracing application, which is not very privacy-friendly: https://www.bbc.com/news/world-asia-55541001

    We have added a section starting with “Some countries were successful in their response to COVID-19. Singapore, South Korea, and China (after the initial first wave) once received high praise for their…..” and have cited the BBC work.

  1. The sentences at the top of page 4 seem to be repetitions of what has already been written at the bottom of page 3.

    Removed the first instance of that statement.

  2. I think all the figures should be edited by removing at least the decorations, to focus only on the essential.

    I don’t know what decoration you mean as the figures do not have any! They are screenshots of applications!

  1. Page 9, at the bottom of the page, there is an "8." What is it?
    I have removed the incorrect reference to figure 8.

  2. The conclusion highlighting the advantages of the decentralized version is a bit lapidary and lacks justification in light of everything else written in the article. A summary of the attacks in a table would be welcome.
    Actually the entire paper (i.e. LS and the experiments) justifies this as we have comprehensively discusses. The attacks or possible ways to leak data is explicit there under the “Framework Evaluation” Section. Each vulnerability/attack/problem is investigated in a paragraph.

  3. The last sentence should be qualified with respect to reference [23] which is a combination of centralized and decentralized contact tracing.
    Yah, we agree with this but this is a future work.

========================================

Reviewer 2 Report

This study examined contract centralized and decentralized contact tracing systems in Europe from a threat perspective to design a framework that enables privacy and security for contact tracing applications. The results would be of interest to developer of these applications.

The paper is well written, with a strong background section. The researchers provided detailed descriptions of the technical security and privacy issues prevalent in these systems. The paper proposed a validated framework that is very novel.

A comment:

RPI: I am not finding where this acronym was defined

Author Response

Reviewer 2:
Firstly, we appreciate the constructive feedback. We have done our best to accommodate them into the manuscript and those we could not, we have provided a reason/explanation. 

1.     This study examined contract centralized and decentralized contact tracing systems in Europe from a threat perspective to design a framework that enables privacy and security for contact tracing applications. The results would be of interest to developer of these applications.
Thank you. We are happy you find the results useful.

2.     The paper is well written, with a strong background section. The researchers provided detailed descriptions of the technical security and privacy issues prevalent in these systems. The paper proposed a validated framework that is very novel.
We appreciate the time you spent here and yes we have a comprehensive framework that is tested/validated.

3.     A comment: RPI: I am not finding where this acronym was defined
It stands for Rolling Proximity Identifier and it is spelled out in Table 3: Questions to evaluate implementations.